# Study of the Influence of Magnetite Nanoparticles Supported on Thermally Reduced Graphene Oxide as Filler on the Mechanical and Magnetic Properties of Polypropylene and Polylactic Acid Nanocomposites

**DOI:** 10.3390/polym13101635

**Published:** 2021-05-18

**Authors:** Benjamin Constant-Mandiola, Héctor Aguilar-Bolados, Julian Geshev, Raul Quíjada

**Affiliations:** 1Facultad de Ciencias Físicas y Matemáticas, Universidad de Chile, Beauchef 851, Santiago 8370456, Chile; benjamin.constant@ing.uchile.cl; 2Departamento de Polímeros, Facultad de Ciencias Químicas, Universidad de Concepción, Concepción 3349001, Chile; haguilar@udec.cl; 3Instituto de Física, URFGS, Porto Alegre 91501-970, Brazil; julian@if.ufrgs.br

**Keywords:** nanocomposites, magnetic properties, thermally reduced graphene oxide magnetite

## Abstract

A study addressed to develop new recyclable and/or biodegradable magnetic polymeric materials is reported. The selected matrices were polypropylene (PP) and poly (lactic acid) (PLA). As known, PP corresponds to a non-polar homo-chain polymer and a commodity, while PLA is a biodegradable polar hetero-chain polymer. To obtain the magnetic nanocomposites, magnetite supported on thermally reduced graphene oxide (TrGO:Fe_3_O_4_ nanomaterial) to these polymer matrices was added. The TrGO:Fe_3_O_4_ nanomaterials were obtained by a co-precipitation method using two types of TrGO obtained by the reduction at 600 °C and 1000 °C of graphite oxide. Two ratios of 2.5:1 and 9.6:1 of the magnetite precursor (FeCl_3_) and TrGO were used to produce these nanomaterials. Consequently, four types of nanomaterials were obtained and characterized. Nanocomposites were obtained using these nanomaterials as filler by melt mixer method in polypropylene (PP) or polylactic acid (PLA) matrix, the filler contents were 3, 5, and 7 wt.%. Results showed that TrGO_600_-based nanomaterials presented higher coercivity (Hc = 8.5 Oe) at 9.6:1 ratio than TrGO_1000_-based nanomaterials (Hc = 4.2 Oe). PLA and PP nanocomposites containing 7 wt.% of filler presented coercivity of 3.7 and 5.3 Oe, respectively. Theoretical models were used to analyze some relevant experimental results of the nanocomposites such as mechanical and magnetic properties.

## 1. Introduction

The preparation of polymeric nanocomposites based on carbon nanomaterials is an excellent alternative to obtain polymeric materials with enhanced mechanical or electrical properties [1,2]. This is because carbon nanomaterials impart mechanical resistance and electrical conductivity to the soft and electrically insulating polymeric matrix [3,4,5]. As known, one of the most widely used nanomaterials as fillers is graphene, due to its diversity of properties, such as its high surface area of 2630 m^2^·g^−1^ and high mechanical properties of 1300 GPa [6]. In this respect, the high surface area of graphene allows for use in a wide range of multiple purposes, such as elastomers, thermoplastic, sustainable glassy nanocomposites [7,8,9] as well as hydrogel-based nanocomposites for environmental remediation, heavy metal absorption, and waste treatment [10,11,12]. Moreover, the high surface area of graphene allows hosting on its surface different nanomaterials which specific properties, such as magnetic properties [13]. 

There are several approaches to obtain graphene materials, one of which considers the thermal, photo, or chemical reduction of graphene oxide (GO) [14,15,16,17,18,19]. Using a modified Hummers’ method, GO can be obtained from graphite. However, the residual presence of functional groups present in GO gives it some differences compared with pristine graphene [20]. The thermal reduction of GO is considered one of the highest efficiency and least polluting procedures for obtaining graphene materials [14]. The heat used to reduce GO not only favors total or partial elimination of oxygen-functional groups present in the GO but also allows the exfoliation of the graphene sheets to take place [21,22]. The exfoliation favors the drastic increase of the surface area of these types of materials. Consequently, the partial functionalization with oxygen moieties and the high surface area of these graphene materials opens an interesting approach to explore the synthesis and support of different types of nanoparticles on the surface of graphene. 

On the other hand, magnetite is an iron oxide, Fe_3_O_4_, which stands out by presenting magnetic properties [23]. Various methods allow the synthesis of magnetite: electrochemistry [24], sol-gel [25], and chemical coprecipitation [26], the latter being more widely used due to its easy scaling up to the industrial level. The coprecipitation method of magnetite is carried out by using as precursors basic aqueous solutions of FeCl_3_ and FeCl_2_, as shown in Equation (1) [27].
(1)$${\mathrm{Fe}}^{2+}+2{\mathrm{Fe}}^{3+}+8{\mathrm{OH}}^{-}\overset{H_{2}O/{\mathrm{NH}}_{3}}{\to }{\mathrm{Fe}}_{3}O_{4}+H_{2}O$$

The synthesis of magnetite can also be performed on nanomaterial surfaces that present a large surface area such as graphene materials. The presence of functional groups favors the nucleation of magnetite on their surface, obtaining a material with magnetic properties [28]. 

On the other hand, the pollution generated by the α-olefin-based thermoplastics polymers has promoted new research with more sustainable polymers such as polylactic acid (PLA), which can be degraded by external agents such as humidity, UV light, and composting, reducing the adverse impact on the environment [29]. However, it is also necessary to find alternatives to recycle nondegradable polymers such as PP and facilitate this process. Obtaining nanocomposites with magnetic properties allows separating and collecting these materials from disposal polymer batches faster using an external magnetic field, facilitating their recycling. The dispersion of the nanoparticles in the polymers depends as much on the nanoparticle-polymer interaction forces as well as on the nanoparticle-nanoparticle interaction forces [30]. Nanoparticles with a higher polarity due to the presence of functional groups or groups with unpaired electrons will have a higher affinity with a polar polymer matrix such as PLA and PA [31]. In the case of nonpolar nanoparticles, they will have a higher affinity with a nonpolar polymer matrix such as PP [5].

Herein, two thermally reduced graphene oxides at 600 and 1000 °C that have different contents of oxygen functional groups were used as supports of magnetite synthesized by a coprecipitation method. The structure, morphology, and magnetic properties of these supported-magnetite nanoparticles on TrGO were studied. Besides, the influence of these hybrid nanomaterials on the properties of the thermoplastic polymer matrices, namely polypropylene and polylactic acid were studied. These matrices were selected because they are polymers with importance in the industry of thermoplastic polymers. For instance, PP is extensively used in household and industrial applications, consequently it is considered a commodity [32]. Conversely, PLA is far less used for manufacturing low price goods, the PLA’s importance stems from its biodegradable character and possesses prospective applications [33]. It is important to mention that PP corresponds to a non-polar homo-chain polymer, while PLA is a polar hetero-chain polymer. As known, nowadays, there is a significant interest in using biodegradable polymers in the thermoplastic industry. However, in both cases, the study of their preparation and magnetic properties could provide information for understanding the behavior of these nanocomposites. 

## 2. Materials and Methods

### 2.1. Materials

Graphite, sulfuric acid (H_2_SO_4_, 98%), potassium permanganate (KMnO_4_, ≥99.0%), hydrochloric acid (HCl, 37%), sodium nitrate (NaNO_3_, 99.0%), iron(III) chloride hexahydrate (FeCl_3_·6H_2_O, 97%), iron(II) chloride tetrahydrate (FeCl_2_·4H_2_O, 98%) and ammonia solution (NH_3_, 25%) were supplied by Merck (Kenilworth, NJ, USA). Polypropylene (PP) PH 2621, density = 905 kg·m^−3^, molecular weight = 195 Kg·mol^−1^, melt flow index = 27 g/10 min (2.16 kg, 230 °C) supplied by Petroquim S.A. (Santiago, Chile). Polylactic acid (PLA) 4032D, density = 1240 kg·m^−3^, molecular weight = 107.3 Kg·mol^−1^, melt flow index = 7 (2.16 kg, 230 °C), supplied by NatureWorks (Minnetonka, Minneapolis, MN, USA). All materials were used as received. 

### 2.2. Synthesis of TrGO/Fe_3_O_4_ Nanoparticles

#### 2.2.1. Graphene Oxide Synthesis and Thermally Reduction Process

The graphene oxide (GO) was obtained by a modified Hummers method, which considers the addition of 15 g of graphite and 7.5 g of NaNO_3_ dispersed in 375 mL of sulfuric acid under constant stirring at 0 °C. The suspension was stirred for 30 min, and then 45 g of KMnO_4_ was added slowly in small portions over a period of one hour. Then the suspension was mixed to room temperature and was left to react for 45 min. The mixture was poured carefully into 750 mL of distilled water, and 675 mL of H_2_O_2_ (5 vol%), added with the purpose of removing the excess of KMnO_4_. The reaction product was washed with HCl (26%) and was filtered several times until the supernatant achieved a pH *ca*. 7. Finally, the GO was dried at 80 °C for 12 h [34]. 

For the thermally reduced graphene oxide (TrGO) synthesis, GO was previously dried for 6 h at 80 °C under vacuum. A small portion of GO was introduced into a quartz reactor, which was sealed and purged using nitrogen gas. The reactor was slowly heated up 600 °C or 1000 °C and the reactor was left for 30 s at the working temperature. The reactor was then allowed to cool down to room temperature. The thermal shock is the prime requirement to achieve exfoliation and reduction of graphene sheet functional groups [14].

#### 2.2.2. TrGO/Magnetite Nanomaterial Synthesis

The TrGO/magnetite nanomaterial was synthesized by coprecipitation of FeCl_3_·6H_2_O and FeCl_2_·4H_2_O in the presence of TrGO_600_ or TrGO_1000_. The aqueous solutions of ferric chloride and ferrous chloride were prepared in a 2:1 mole ratio. Two solutions with different concentrations were used in order to obtain materials with different FeCl_3_ and TrGO_600_ or TrGO_1000_ weight ratios of 2.5:1 and 9.6:1 FeCl_3_:TrGO, resulting in four samples of nanomaterials: (Mb_600_, Mb_1000_, Ma_600_, and Ma_1000_). The preparation of Mb_600_ and Mb_1000_ consisted of the addition of 50 mg of TrGO_600_ (for Mb_600_) or TrGO_1000_ (for Mb_1000_) to 50 mL of DI water. These suspensions were ultrasonicated for 1 h, and then 50 mL of FeCl_3_ (125 mg) and FeCl_2_ (49 mg) solution in DI water was added at 80 °C. Similarly, the preparation of Ma_600_ and Ma_1000_ considered the addition of 50 mg of TrGO_600_ (for Ma_600_) or TrGO_600_ (for Ma_1000_) in 50 mL of DI water was ultrasonicated for 1 h. Then 50 mL of FeCl_3_ (480 mg) and FeCl_2_ (188 mg) solutions were added at 80 °C [28,35,36]. 

After mixing the ferric chloride and ferrous chloride solutions with the TrGO solution, a 25% ammonia solution was added, increasing the pH to 10 at 80 °C. This process was carried out under a nitrogen gas flow. The resulting suspensions were stirred for 30 min. The suspensions were cooled at 0 °C in an ice bath, the nanomaterial was separated using a magnet and was washed three times with DI water, and finally vacuum-dried at 60 °C for 2 h.

### 2.3. Characterization of Nanoparticles 

The different graphene materials were characterized by elemental analysis carried out using a Perkin Elmer MCHNSO/2400 analyzer (Waltham, MA, USA). These materials also were characterized by powder X-ray diffraction (XRD) and were recorded using a Siemens D-5000 wide-angle XRD spectrometer (München, Germany) with Cu Kα radiation, operating at 40 kV and 30 mA. The Raman spectroscopy was performed in an inVia Renishaw Raman spectrometer (Wotton-Under-Edge, UK) equipped with a 532 nm laser at a power of 10 mW. The specific surface areas were analyzed in a Nova Station A analyzer Quantachrome instruments, Anton Paar (Ashland, Wilmington, DE, USA), through the determination of the multi-point Brunauer-Emmett-Teller (BET) plot using nitrogen gas, all curves presented an R^2^ higher than 0.999 [37,38]. Scanning electron microscopy (SEM) studies were performed in an EVO MA10 Carl Zeiss microscope (Jena, Germany) operating at 13.4 kV. 

The magnetic properties of TrGO:Fe_3_O_4_ (powder) were measured as a function of the applied magnetic field using an EZ29MicroSense vibrating magnetometer (VSM) with a maximum applied magnetic field of 20 KOe. The hysteresis of the magnetization was obtained by varying H between +20 KOe and −20 KOe at room temperature.

### 2.4. Preparation of Nanocomposites

The nanocomposites were prepared by melt mixing in a Brabender Plasticorder double screw mixer (Nordrhein-Westfalen, Germany). The nanocomposites based on PP and PLA nanocomposites were mixed at 190 and 200 °C for 10 min using a rotor speed of 110 RPM. The filler content in the nanocomposites varied over a range of 3–7% by weight. The PP, PLA, and TrGO:Fe_3_O_4_ nanomaterial were dried at 80 °C for 8 h prior to being mixed. Then, the nanocomposites were processed in an HP hydraulic press (model D-50) with a heating system, and a water-cooling system. The sample thickness was 1.0 nm for tensile and magnetic tests.

### 2.5. Characterization of Nanocomposites

The mechanical properties were determined by tensile-strain tests according to the ASTM D638 standard, at a rate of 25 mm·min^−1^ at room temperature using an Instron Universal Testing System model 3382, (Norwood, MA, USA). Three specimens were tested for each sample and the average was determined to obtain a representative data dispersion. 

The magnetic properties of nanocomposite (films) were measured as a function of the applied magnetic field using an EZ29MicroSense vibrating magnetometer (VSM) (Lowell, MA, USA) with a maximum applied magnetic field of 20 KOe. The hysteresis of the magnetization was obtained by varying H between +20 KOe and −20 KOe at room temperature.

### 2.6. Theoretical Models

#### 2.6.1. Halpin-Tsai Model

As known, one of the important aspects that influence the mechanical properties of nanocomposites (Young moduli) is the aspect ratio and filler/polymer matrix adhesion. Halpin Tsai model provides information about these features by considering the theoretical moduli of the filler, the matrix, and the experimental moduli of the different nanocomposites. Equations (2) and (3) present the de Halpin Tsai approach [39,40].
E = E_M_∗(1 + ƞ2αV_f_)/(1 − ƞV_f_)(2)
Ƞ = (E_f_/E_M_ − 1)/(E_f_/E_M_ + 2α)(3)
where E_M_ is the young modulus of the polymer matrix, E_f_ corresponds to the Young modulus of the filler, α is the aspect ratios of the filler nanoparticles, which also considers their geometry and distribution, and V_f_ corresponds to the filler volume fraction. 

#### 2.6.2. Langevin Model

In order to understand the influence of the magnetic properties of the filler on the matrix polymerics, the Langevin model was used for determining the magnetic moment and nanoparticle size [41]. The Langevin equation, as shown in Equation (4), considers that each monodomain has magnetic moments randomly oriented, which interact with the external magnetic field.
M = Nµ∗(coth(µµ_o_H/k_b_T) − k_b_T/µµ_o_H)(4)
where µ is the magnetic moment, µ_o_ is the vacuum permittivity, H the external magnetic field, k_b_ corresponds to the Boltzman constant, T is the temperature and N is the Avogadro number.

## 3. Results and Discussion

### 3.1. Characterization of Thermally Reduced Graphene Oxide

Figure 1a shows the normalized XRD patterns of graphite, GO, TrGO_600_, and TrGO_1000_. Graphite presents an intense diffraction peak at 2θ = 26.3°, corresponding to the (002) plane which is associated with the interlayer spacing between graphene layers (0.338 nm). Once the graphite was oxidized, this peak shifted to 2θ = 12.6°, which indicates an increase in the interlayer distance of 0.702 nm. The increase of the interlayer distance is associated with the incorporation of oxygenated functional groups to the graphitic structure. The thermal reduction process of GO at 600 or 1000 °C produced a shift of this peak to 2θ = 25.1° and 2θ = 26.02°, corresponding to the interlayer spacing of 0.35 nm and 0.34 nm, respectively, as seen in Table 1. Table 1 also presents the chemical composition and the BET surface area of GO, TrGO_600_, and TrGO_1000_. In fact, the oxygen contents of TrGO_600_ and TrGO_1000_ are lower compared to that of GO (43.6%), and they are 15.7 and 8.7%, respectively. The lowest content achieved in TrGO_1000_ is because a higher temperature favors a more complete reduction. Concerning the BET surface area results of the nanomaterials (Table 1), it is seen that the surface area increases from 69 m^2^·g^−1^ for GO to 304 and 267 m^2^·g^−1^ for TrGO_600_ and TrGO_1000_, respectively. The increase of the surface area proves the effectiveness of the exfoliation of GO by a thermal reduction process.

### 3.2. Characterization of TrGO/Fe_3_O_4_ Nanomaterials

Figure 1b shows the XRD patterns of Mb_600_, Mb_1000_, Ma_600_, and Ma_1000_ magnetite (Described in Methodology 2.1.3), all of them showing six peaks at the 2θ angles of 30.2°, 35.4°, 43.3°, 53.8°, 57.2°, and 62.7°. The peak seen at 35.4° presents the highest intensity, which is characteristic of the magnetite-based structures. This peak probably corresponds to the (311) magnetite plane with a face-centered cubic arrangement. The diffraction pattern of the TrGO:Fe_3_O_4_ showed the characteristic peaks of Fe_3_O_4_, indicating the successful deposition of Fe_3_O_4_ on the TrGO surface. Moreover, the peak associated with the (002) plane (2θ = 26.4°) of TrGO was observed only in the Mb_600_ and Mb_1000_ samples as seen in Figure 1a, indicating that the TrGO surface probably has a lower magnetite content.

The crystal size was estimated using the Debye-Scherrer equation, and it was possible to determine that Fe_3_O_4_ had a 19.4 nm crystal size. Similar values of (311) crystal size of the Fe_3_O_4_-based structure deposited on TrGO are shown by samples Mb_600_ (16.5 nm), Mb_1000_ (12.2 nm), Ma_1000_ (16.2 nm), and Ma_600_ (19.2 nm). The crystal size of magnetite particles supported on TrGO tends to be comparable to the values reported by Baumgartner et al. [42], who used similar pH, temperature, and reaction time conditions. In addition, A. Zubir et al. demonstrated the formation of a mixed oxide of Fe(II) and Fe(III), namely as Fe_3_O_4_ supported on graphene oxide, using a similar method of coprecipitation. This was supported by using X-ray photoelectron spectroscopy for the characterization of these nanomaterials, wherein the disappearing of Fe 2p_3/2_ photoelectron line indicated the existence of the Fe(II) and Fe(III) species [43]. 

Figure 2a,b shows the Raman spectra of nanomaterials of Mb_600_, Mb_1000_, Ma_600_, and Ma_1000_. As expected, *D* and *G* bands characteristic of graphene materials are seen (Figure 2a). These bands appear at ca. 1350 cm^−1^ and 1580 cm^−1^, respectively [44,45]. It is seen that the sample corresponding to Ma_1000_ presents a Raman redshift, which is associated with the interaction of graphene layers with nanoparticles [46]. This suggests the occurrence of interaction between the Fe-based structure and the graphene oxygen moieties, favoring the nucleation structure on the graphene structure. On the other hand, Figure 2b shows the region of the Raman spectra between 150 cm^−1^ and 1000 cm^−1^, presented to study the contribution of magnetite absorption bands. Bands are seen at ca. 212 cm^−1^, 274 cm^−1^, and 385 cm^−1^ [47].

Regarding the BET analysis, samples Mb_600_, Mb_1000_, Ma_600_, and Ma_1000_ presented surface areas of 143.7, 166.8, 111.2, and 123.8 m^2^·g^−1^, respectively. It can therefore be inferred that these decreases depend on the type of graphene used. TrGO:Fe_3_O_4_ nanomaterials using TrGO_600_ as support present greater decreases of their surface areas compared to nanomaterials using TrGO_1000_ as support. TrGO_600_ presented more oxygen functional groups on their surface than TrGO_1000_, probably favoring the synthesis of magnetite on the surface of TrGO_600_. In samples with many magnetite nanoparticles, graphene layers are hindered, and their large available surface decreases. 

Figure 3 shows SEM images of TrGO:Fe_3_O_4_ nanomaterials. In the case of nanomaterials of Mb_600_ and Mb_1000_ (Figure 3a,b), small magnetite particles are seen on the surface of the graphene flakes and tend to present an amorphous structure. In the case of Ma_600_ and Ma_1000_ nanomaterials (Figure 3c,d), they present a highly compacted morphology and heterogeneous distribution of nanoparticle sizes on the surface. The coprecipitation method does not produce homogeneous magnetite particle sizes, so it is possible to find broad size dispersion of magnetite particles on the surface of TrGO [42]. The appearance of the TrGO:Fe_3_O_4_ nanomaterials indicates that the Fe_3_O_4_ was deposited on the TrGO surfaces, hiding the flakes of this carbon-based nanomaterial. Consequently, the nanomaterials have heterogeneous character; namely, flakes of TrGO, covered by magnetite (Fe_3_O_4_).

### 3.3. Magnetic Properties of Nanomaterials

Figure 4 shows the cycles of magnetic hysteresis curves of TrGO:Fe_3_O_4_ nanomaterials, where the superparamagnetic behavior of these samples is seen. Nanomaterials Mb_600_ and Mb_1000_ presented magnetic saturation (M_s_) of 37.5 and 39.9 emu·g^−1^, and coercivity (H_c_) of 0.373 Oe and 0.023 Oe, respectively. On the other hand, nanomaterials Ma_600_ and Ma_1000_ presented M_s_ values of 55.2 and 63.01 emu·g^−1^, respectively, and the H_c_ were 8.509 and 4.219 Oe, respectively. The tendency to increase M_s_ and coercivity H_c_, which can be attributed to the higher Fe amount used to synthesize the Ma_600_ and Ma_1000_ nanomaterials. This can be inferred from the results of X-ray diffraction analysis and Raman spectroscopy. TrGO_600_-based nanomaterials have lower magnetic susceptibility but greater coercivity compared to TrGO_1000_-based nanomaterials. These results suggest that the coercivity and magnetic susceptibility depend on the crystal size of Fe_3_O_4_. 

### 3.4. Properties of PP and PLA of Nanocomposites

#### 3.4.1. Mechanical Properties

Figure 5 shows the Young’s modulus of the nanocomposites. In the case of PP-TrGO:Fe_3_O_4_ nanocomposites (Figure 5a), it is seen that their Young’s moduli have different behaviors depending on the type of filler used and their concentrations. PP nanocomposites containing 5% Ma_600_ as filler presented a 21% increase of Young’s modulus. When the filler content was increased to 7%, the Ma_1000_-based nanocomposites presented a 20% increase, while the Ma_600_-based nanocomposites had a decrease of their Young’s modulus. The different behaviors of the nanomaterials’ Young’s modulus are attributed to the affinity that nanomaterials have with the PP matrix. A nonpolar nanomaterial has a better affinity with the nonpolar polymer matrix, favoring the obtaining of homogeneous filler dispersion on the polymer matrix. Conversely, nanomaterials with a polar nature have less affinity with a polar polymer matrix such as PP; consequently, filler agglomerations in the polymer matrix take place. Mb_1000_ presented a higher affinity with the PP matrix, generating increased mechanical properties. This can be attributed to the fact that Mb nanocomposites have a lower content of magnetite, which has a polar nature. This probably favors the achievement of a homogeneous dispersion of filler in the PP matrix and an improved filler/polymer interfacial adhesion. Furthermore, the nanoparticle size favored the adhesion of the nanomaterial with the PP matrix. This is reflected in an increase of Young’s modulus [48]. However, when the filler content increased to 7%, a drastic reduction of mechanical properties took place.

In the case of PLA-TrGO:Fe_3_O_4_ nanocomposites (Figure 5b), it is seen that Young’s moduli depend on the filler content and its nature. PLA composites containing 5% filler of Ma_1000_ exhibited an 11% increase in Young’s modulus. Furthermore, Young’s modulus decreases 28% using a 7% filler of the same nanomaterials. There are two factors that likely can explain the behavior of the mechanical properties of PLA-TrGO:Fe_3_O_4_, namely reprocessing of nanomaterials and agglomeration in the polymer matrix. Probably, the processing of PLA at temperatures close to the melting point, 160–180 °C, generates a partial degradation of the polymer chains, as it was reported in the study of Carrasco et al. [49]. The PLA polymer chains start degrading by releasing methyl groups, causing a decrease in molecular weight, and consequently generating changes in the crystallinity of the polymer. This causes a decrease in the mechanical properties, which are linked to the polymer’s intermolecular forces [49,50]. In addition, the unexpected decrease of Young’s moduli of the different nanocomposites also can be related to the heterogeneous nature of the fillers. Magnetite and TrGO likely present weak interactions, and this fact imparts defects in the nanocomposites. As a result, the stiffness of the nanocomposites will decrease in the presence of TrGO:Fe_3_O_4_, and the brittleness of the nanocomposites will be higher than the neat polymer matrices. This suggests that the use of a binding agent or organic surface modification can be a strategy to increase the interaction between fillers and polymer matrix and to avoid the decrease of the mechanical properties [51,52,53,54,55]. 

Figure 6 shows the Halpin Tsai-determined theoretical and experimental young moduli as a function of the filler content of nanocomposites based on PP and PLA, respectively. In Figure 6a It is possible to observe the nanocomposites containing Mb_1000_ and Ma_1000_ as filler. As seen in this figure, the Halpin Tsai model tends to converge with experimental data, when the aspect ratio is α = 20 for PPMa_1000_. On the other hand, the estimated aspect ratio of the nanocomposite PPMb_1000_ is α = 5. However, the experimental results show a poor correlation with the theoretical data, where nanocomposite filled with 7% of Mb_1000_. This may be due to the fact that there are interactions between the nanoparticles that can favor or decrease the adhesion of the nanoparticle with the polymer matrix [56]. 

On the other hand, in Figure 6b, the data of Young’s modulus of PLA with filling of Mb_600_ and Ma_600_ with the Halpin Tsai model are shown as a function of the filler content. It is possible to observe that the theoretical data do not fit with the experimental results when α = 1 is used. This suggests that the adhesion of the PLA with nanofiller is low. Consequently, the experimental Young moduli of nanocomposite are lower than the expected values [57].

#### 3.4.2. Magnetic Properties

Figure 7 shows the magnetic hysteresis cycles of PP and PLA nanocomposites with nanomaterials of Mb_600_ and Mb_1000_ as a filler with a content of 3, 5, and 7 wt.%. In the case of PP nanocomposites (Figure 7a), it was found M_s_ value of 3.21 emu·g^−1^ and a H_c_ of 0.24 Oe using Mb_600_ with 7 wt.% filler. These data agree with the mechanical properties found for this nanocomposite with 7 wt.% filler of where Mb_600_ presents a decrease of these properties due to the agglomerations of the nanomaterials, favoring the contact between the nanoparticles, allowing that a higher magnetic saturation can be achieved in the nanocomposite.

In the case of PLA nanocomposites, they present a high M_s_ of 3.2 emu·g^−1^ using Mb_1000_ with 7% filler (Figure 7b). PLA had a weak interaction with the Mb_1000_ nanoparticles when a 7% by weight filler was used, increasing the formation of agglomerations due to the interactions between nanomaterials, increasing the magnetic nanocomposite’s susceptibility. On the other hand, the coercivity of the PLA nanocomposites is higher than that of the nanoparticles, reaching 0.810 Oe with Mb_600_ at 7% filler, more than 200% of coercivity of the nanoparticle. 

Figure 8 shows magnetic hysteresis cycles of PP and PLA nanocomposites with nanomaterials Ma_600_ and Ma_1000_ filler. PP nanocomposite magnetic properties showed a high M_s_ of 4.71 emu·g^−1^ using Ma_1000_ at 7% filler content and a high H_c_ of 5.35 using Ma_600_ at 7% filler (Figure 8a). Ma_1000_ presents a greater magnetic susceptibility than Ma_600_, so the PP nanocomposites based on this nanomaterial presented a higher magnetic susceptibility, and the contact between the nanoparticles with the polymer matrix likely favors the increase of Ms. In the case of PLA nanocomposites, they present a similar behavior as PP nanocomposites, while when using a 7 wt.% filler of Ma_1000_ a Ms of 4.8 emu·g^−1^ and a high H_c_ of 3.97 Oe using 5 wt.% Ma_1000_ as a filler (Figure 8b). Compared to PP nanocomposites, a difference in magnetic properties is seen due to interactions between the PLA matrix and the magnetite nuclei of the nanomaterials.

Figure 9 shows results of Langevin model for the nanocomposite, where is observed that the PP-7% Ma_600_ nanocomposites curve (Figure 9a) fits with the Langevin model and the estimated particle size is 68 nm and a magnetic moment of 2747 µ_b_. On the other hand, it is observed that the experimental curve of the nanocomposite based on 7% Ma_1000_ also adjusts with the Langevin model. The calculated magnetic moment is 3113 µ_b_ and the particle size is 72 nm. It is possible to note that in both cases of PLA nanocomposites presents higher magnetic saturation than those of PP-based nanocomposites, which indicates a higher magnetic moment [58]. On the other hand, there is a correlation of particle size obtained in this theoretical approach with the results presented in the XRD section.

## 4. Conclusions

PP and PLA nanocomposites with TrGO:Fe_3_O_4_ as filler were prepared by melt mixing. TrGO_600_ has a greater number of functional groups, a larger surface area, and lower content of structural defects than TrGO_1000_, favoring the nucleation of ferric oxides (magnetite) in its surface, obtaining an average particle size of 18.1 nm and presenting a higher coercivity of 8.5 Oe. Nanomaterials of Ma_600_ and Ma_1000_ (high concentration of Fe) have higher magnetic susceptibility and coercivity than Mb_600_ and Mb_1000_. This was attributed to increased magnetite nucleation in graphene structures, covering their entire surface. The highest coercivity (5.3 Oe) observed was shown for PP nanocomposite using a 7% filler content of Ma600. In the case of PLA nanocomposite, the highest coercivity value was 3.9 Oe using 7% filler content of Ma1000. Langevin model was used in magnetic properties to estimate the momentum magnetic and nanoparticle size. All models fit magnetic properties, where PLA with 7% of Ma600 presented high momentum magnetic of 3113 µ_b_ and the particle size of 72 nm. On the other hand, the Halpin Tsai model was used to predict the aspect ratio of the nanocomposite by using the mechanical properties, only for PP with Ma1000 showed a good fit. While PLA nanocomposite did not fit properly. The decrease of the mechanical properties of matrices as a result of the addition of the filler can be related to the heterogeneous nature of the fillers based on magnetite and TrGO. The weak interaction between the magnetite and TrGO imparts defects in the nanocomposites. The results of this research provide evidence about the magnetic properties of nanocomposites based on recyclable or biodegradable thermoplastic polymers and shed light on the challenges to obtain magnetic polymer nanocomposites with proper mechanical properties performance.

## Figures and Tables

**Figure 1 polymers-13-01635-f001:**
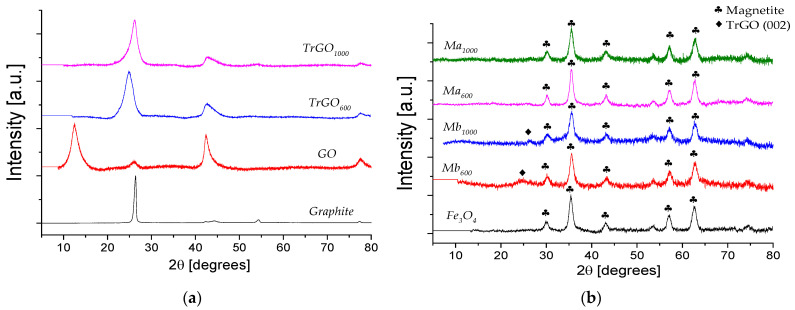
X-ray diffraction patterns of: (**a**) graphite, GO, TrGO_600_, and TrGO_1000_ and (**b**) Fe_3_O_4_, Mb_600_, Mb_1000_, Ma_600_, and Ma_1000_.

**Figure 2 polymers-13-01635-f002:**
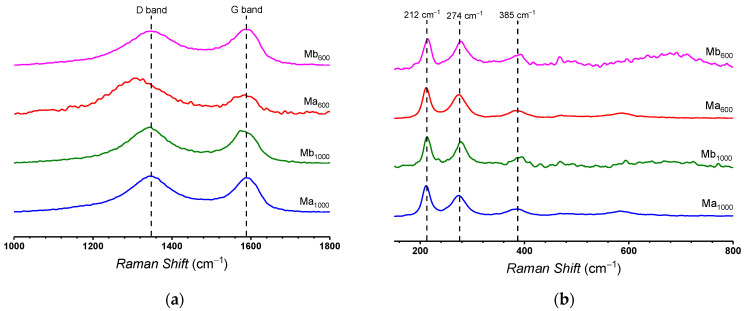
Raman spectra of TrGO: (**a**) and (**b**) TrGO/Fe_3_O_4_.

**Figure 3 polymers-13-01635-f003:**
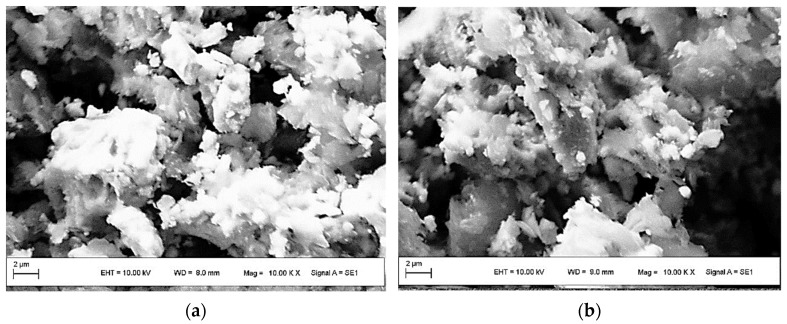
SEM images of TrGO:Fe_3_O_4_ nanomaterials: (**a**) Mb_600_, (**b**) Mb_1000_, (**c**) Ma_600_, and (**d**) Ma_1000_.

**Figure 4 polymers-13-01635-f004:**
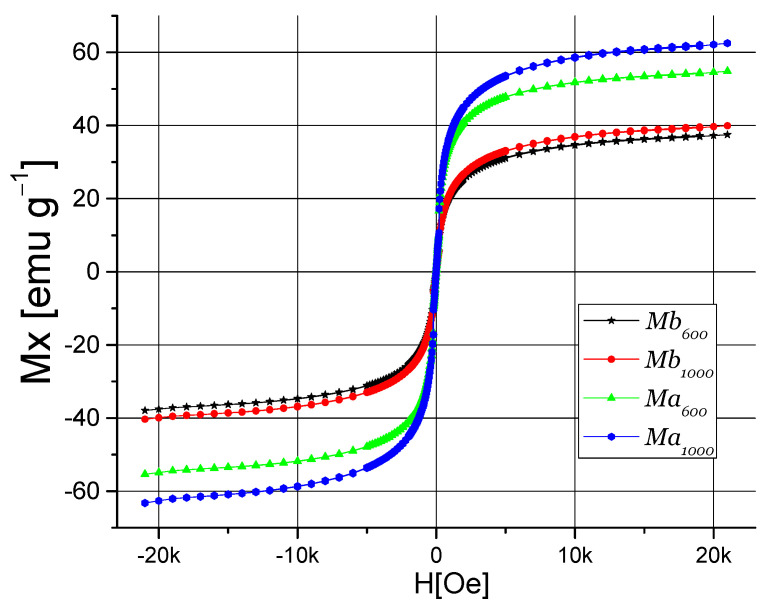
Cycles of Hysteresis magnetic curve of TrGO:Fe_3_O_4_ powder: Mb_600_, Mb_1000_, Ma_600_, and Ma_1000_.

**Figure 5 polymers-13-01635-f005:**
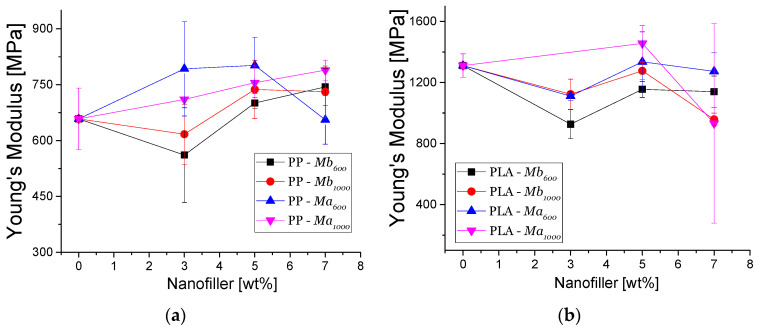
Young’s modulus of nanocomposites: (**a**) PP-TrGO:Fe_3_O_4_; (**b**) PLA-TrGO:Fe_3_O_4_.

**Figure 6 polymers-13-01635-f006:**
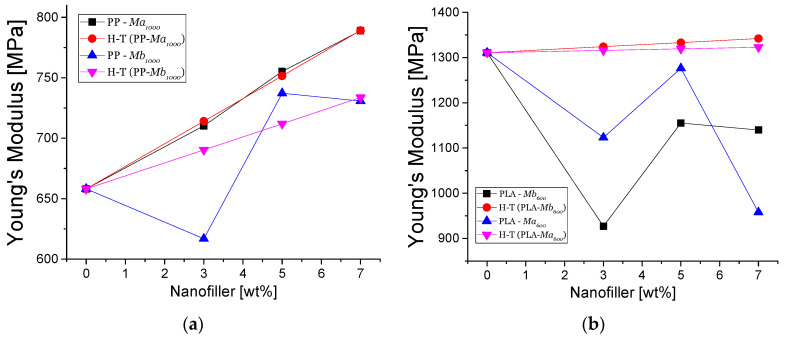
Halpin Tsai model for nanocomposites of PP-TrGO:Fe_3_O_4_ (**a**) and PLA-TrGO:Fe_3_O_4_ (**b**).

**Figure 7 polymers-13-01635-f007:**
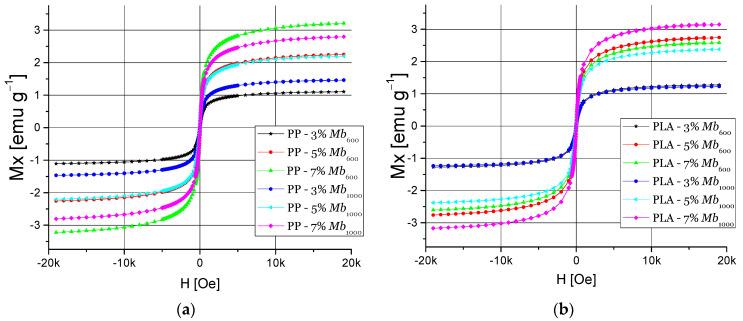
Magnetic hysteresis cycles: (**a**) PP:Mb_600_ and Mb_1000_ and (**b**) PLA:Mb_600_ and Mb_1000_ nanocomposites.

**Figure 8 polymers-13-01635-f008:**
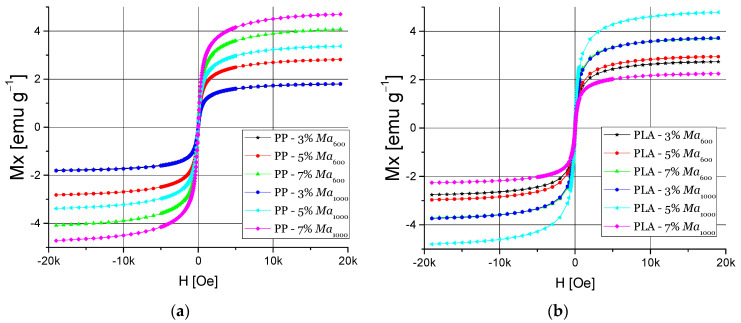
Magnetic hysteresis cycles: (**a**) PP:Ma_600_ and Ma_1000_ and (**b**) PLA:Ma_600_ and Ma_1000_ nanocomposites.

**Figure 9 polymers-13-01635-f009:**
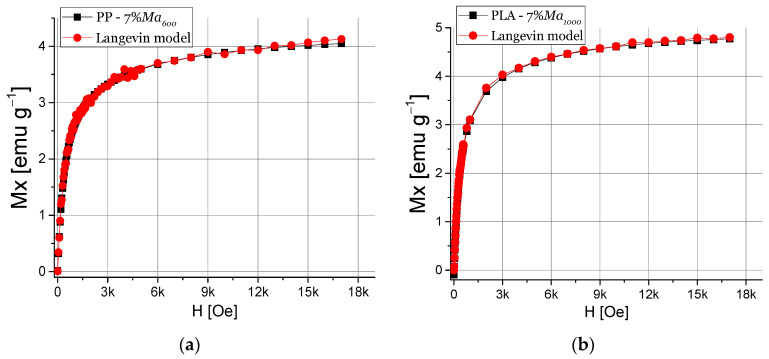
Langevin model for magnetic properties of PP-TrGO:Fe_3_O_4_ (**a**) and PLA-TrGO:Fe_3_O_4_ (**b**).

**Table 1 polymers-13-01635-t001:** Interlayer spacing (d_002_ nm) and crystal size (L_c_ nm), elemental analysis, and BET Surface area of GO, TrGO_600_, and TrGO_1000_.

Sample	XRD		Elemental Analysis				BET Surface Area
	d_002_ nm	L_c_ nm	%C	%H	%N	%O	m^2^·g^−1^
GO	0.702	4.08	54.4	1.9	0.1	43.6	69.7
TrGO_600_	0.350	3.42	83.9	0.3	0.1	15.7	304
TrGO_1000_	0.340	4.01	90.8	0.4	0.1	8.70	267

## Data Availability

Data are contained within the article.

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
