# Peer review of "Study of the Influence of Magnetite Nanoparticles Supported on Thermally Reduced Graphene Oxide as Filler on the Mechanical and Magnetic Properties of Polypropylene and Polylactic Acid Nanocomposites"

_polymers, 2021, doi:10.3390/polym13101635_

Round 1

Reviewer 1 Report

The revised version has well addressed my comments and suggestion. I would like to recommend it in the current version. 

Author Response

Dear Reviewer,

Thank you for your valuable comments and recommendation. 

Best wishes.

Reviewer 2 Report

The authors made all the required changes to the manuscript and I suggest acceptance in its present form

Author Response

Dear Reviewer, 

Thank you very much for your valuable comments. 

Best wishes

Reviewer 3 Report

The authors present the synthesis and characterization of a composite material. The characterization is thorough but the research lacks clear application. The topic fits well the scope of the journal. There are multiple aspects of the manuscript that needs improvement before further consideration.

1, The abstract is too technical. It should be rewritten in a way that it gives a brief introduction to the topic and its importance.

2, What is the effect of the polymer molecular weight on the performance? The authors should elaborate on these important aspects.

3, The Raman spectra in Figure 2 should be annotated for easier and quicker understanding.

4, XPS could be useful to characterize the composite material.

5, Recent natural polymer graphene nanocomposite materials should be acknowledged briefly (10.1016/j.apmt.2020.100878; 10.1016/j.cej.2020.124965; 10.1016/j.eti.2020.100664).

6, The robustness of the prepared composite material should be demonstrated.

7, The purity/grade for all chemicals used in the study should be presented under the experimental section, more specifically the 2.1 Materials.

8, The choice of materials should be justified. The composite material consists of several constituents and their role and the hypothesis behind their selection should be introduced in the manuscript.

9, Both the quotient (“x/y”) and negative exponent (“x y-1”) formats are used in the manuscript for units. Either of them should be used consistently, preferably the negative exponent format.

10, The use of PLA in composite materials should be briefly mentioned (10.1021/acsapm.0c00673; 10.1021/acssuschemeng.9b02516; 10.1021/acsapm.0c00790).

11, How was the BET surface area deduced? The authors should discuss which part of the curve was used to derive the surface area? This is an important aspect that results in significant difference in the BET surface area. Error should be also reported for the BET data.

12, The potential impact of the work should be mentioned under the Conclusion part.

Round 2

Reviewer 3 Report

The authors have done a thorough revision and addressed the comments.

This manuscript is a resubmission of an earlier submission. The following is a list of the peer review reports and author responses from that submission.

Round 1

Reviewer 1 Report

Recheck the first sentence in the introduction section and amend it.

There are very short sentences that should be revised to be suitable for the reader.

Materials section should be separately added before experimental parts with insight on the source and country of each utilized substance.

What is the origin countray for the utilized instruments such as XRD, Raman, SEM, and EZ29MicroSense vibrating magnetometer?.

The data of SEM images should be extensively discussed.

Reviewer 2 Report

Polypropylene (PP) and Polylactic acid (PLA) nanocomposites containing thermally reduced graphene oxide with magnetite (TrGO-Fe3O4) were prepared by melt mixing to study the effect of this filler on the mechanical and magnetic properties. Two different TrGOs were obtained from GO by thermal reduction at 600 °C and 1000 °C aimed at controlling the oxygen functional groups contained in TrGO, and to study the effect of these groups on the nucleation of magnetite nanoparticles.

Results showed that TrGO600-based nanomaterials presented higher coercivity (Hc= 8.5 Oe) at 9.6:1 ratio than TrGO1000-based nanomaterials (Hc= 4.2 Oe) due to the higher oxygen functional group content and specific surface area of TrGO600.

The work has been an extension of the authors work (ref #2 and #3), among others. Since such fillers have been studied before (see: Segregated Hybrid Poly(methyl methacrylate)/Graphene/Magnetite Nanocomposites for Electromagnetic Interference Shielding: Farbod Sharif, Mohammad Arjmand, Aref Abbasi Moud, Uttandaraman Sundararaj, Edward P. L. Roberts, ACS Appl. Mater. Interfaces 2017, 9, 16, 14171–14179, among many others), this work lacks novelty. Most of the conclusions are obvious.

This reviewer cannot recommend this paper for publications in Polymers.

Reviewer 3 Report

The present study reports on study of the influence of magnetite nanoparticles and reduced graphene oxide on the mechanical and magnetic properties of PP and PLA, and a series of experimental results have been carried out to support it. And then the two models have been introduced to verify the properties of the composite. After carefully reading it, the following points are suggested to consider to improve this study.

1. Why the hybrid of TrGO/Fe3O4 is used in this study, it is not clear. Although it is presented that the GO, and Fe3O4 are respectively used in the polymer matrix. TrGO is used to support the Fe3O4 by the functional groups which react with the polymer matrix? How about the interface details between the TrGO and PP/PLA?

2. PP and PLA both are used in figures 6-9, one polymer matrix is better, as the effect of nanofiller on polymer matrix is the same to each other.

3. Images in figure 3 are not clear, the presented morphologies are not useful to support this study.

4. The effect of dispersion of the nanofiller has not been covered in this study. The working principles of effects of TrGO/Fe3O4 on the mechanical and magnetic properties have not been clearly presented, the reasons behind these experimental results, and what the contribution of this study is useful for the readers. These informations are not enough. Please present and discuss on them, and the following references on alignment of nanofiller can be cited for references, i.e. (1) F Carosio , J Kochumalayil, F Cuttica, G Camino, L Berglund. Oriented clay nanopaper from biobased components--mechanisms for superior fire protection properties. ACS Appl Mater Interfaces. 2015, 7(10):5847-56. (2) Haibao Lu, Fei Liang and Jihua Gou. Nanopaper Enabled Shape-Memory Nanocomposite with Vertically Aligned Nickel Nanostrand: Controlled Synthesis and Electrical Actuation. Soft Matter. 2011, 7(16): 7416-7423. (3) Haibao Lu and Wei Min Huang. Synergistic effect of self-assembled carboxylic acid-functionalized carbon nanotubes and carbon fiber for improved electro-activated polymeric shape-memory nanocomposite. Applied Physics Letters. 2013, 102(23): 231910.

Round 2

Reviewer 2 Report

We can't possibly use different polymers as fillers and combine with carbonaceous materials and write a paper on such nanocomposites. They have explored similar strategy in their earlier papers #2 and #3, besides several others that exist with out much rationale.   It is routine work and hence I can't recommend its publication in Polymers. However, if you wish, you can solicit additional opinions.